# Different Effects of the COVID-19 Pandemic on Exercise Indexes and Mood States Based on Sport Types, Exercise Dependency and Individual Characteristics

**DOI:** 10.3390/children8060438

**Published:** 2021-05-24

**Authors:** Alireza Aghababa, Georgian Badicu, Zahra Fathirezaie, Hadi Rohani, Maghsoud Nabilpour, Seyed Hojjat Zamani Sani, Elham Khodadadeh

**Affiliations:** 1Department of Sport Psychology, Sport Sciences Research Institute, Tehran 15879, Iran; alirezaaghababa@yahoo.com; 2Department of Physical Education and Special Motricity, Faculty of Physical Education and Mountain Sports, Transilvania University of Brasov, 500068 Brasov, Romania; georgian.badicu@unitbv.ro; 3Motor Behavior Faculty, Physical Education and Sport Sciences Faculty, University of Tabriz, Tabriz 51666, Iran; hojjatzamani8@gmail.com (S.H.Z.S.); khodadadehelham1995@gmail.com (E.K.); 4Department of Exercise Physiology, Sport Sciences Research Institute, Tehran 15879, Iran; h_rohani7@yahoo.com; 5Department of Sport Physiology, Faculty of Psychology and Educational Sciences, Mohaghegh Ardabili University, Ardabil 56199, Iran; nabilpour@yahoo.com

**Keywords:** exercise indexes, exercise dependency, COVID-19 pandemic, team sports, individual sports

## Abstract

Exercise indexes have been affected by the coronavirus disease 2019 (COVID-19) pandemic and its related restrictions among athletes. In the present study, we investigated the exercise frequency and intensity before and during the COVID-19 pandemic, and also current exercise dependency and mood state among non-contact individual, contact individual, and team sports athletes. A total of 1353 athletes from non-contact individual sports athletes (NCISA), contact individual sports athletes (CISA) and team sport athletes (TSA) participated; 45.4% of them were females that completed a series of self-rating questionnaires covering sociodemographic information, former and current exercise patterns, exercise dependency and mood states. NCISA had less exercise frequency than CISA, both before and during the COVID-19 pandemic, and NCISA had less exercise frequency than TSA during the COVID-19 pandemic. Regarding exercise intensity, CISA had higher scores than NCISA and TSA before the COVID-19 pandemic, and CISA had more exercise intensity than TSA during the COVID-19 pandemic. Frequency and intensity were reduced from before to during the COVID-19 pandemic in the three groups, except for TSA intensity. In addition, positive and negative mood states were correlated with exercise dependency. CISA were more discouraged and vigorous than NCISA and TSA, respectively. For NCISA, CISA, and TSA, ordinal regressions separately showed that adherence to quarantine and exercise dependency were better predictors of exercise indexes. Finally, exercise dependency subscales were different among sports, but it was not in exercise dependency itself. Although the decrease in exercise indexes was noticeable, there was no consistent pattern of change in exercise behavior in all sports. Additionally, during the COVID-19 pandemic, negative moods were predominant among all athletes. The results discussed are based on exercise nonparticipating, sport type, and affect regulation hypothesis.

## 1. Introduction

Psychological and social pressures such as economic problems and illness may always be around us and can lead to some changes in our lifestyle [1,2]. Of course, in addition to stress, the perception of eustress can also have better effects. In fact, what we perceive from external events affects our perception, behavior, and life. The emergence of the coronavirus disease 2019 (COVID-19) is one of these stressors that has greatly affected economic, social, and even personal life. Of course, different strategies have been taken to prevent infection, and some suggestions have been made. In this regard, although the importance of maintaining exercise in all its dimensions, such as physiological or psychological effects, is recommended, the closure of sports centers and the possibility of air pollution in these spaces may have reduced the amount of physical activity (PA) during lockdown [3], and may, in turn, induce numerous health problems such as stress, depression, and anxiety related to the confinement and prolonged periods of inactivity [4]. Recent findings have shown this in various countries [5]; however, a limitation of previous studies was that apparently no distinction was made between individual and team sports athletes. For the following reason, this is critical: compared to individually exercising athletes, it is conceivable that team sports athletes decreased their exercise levels more rigorously when compared to exercise levels before the lockdown, because of social distancing. Additionally, limited training group sizes might have impacted team sports athletes more severely. Thus, the first aim of the present study was to investigate the differences in exercise frequency and intensity of individual and team sports athletes before and during the lockdown.

Decreased exercise in addition to physiologically destructive effects can also have psychological effects, although recent studies have shown a decreased PA during the COVID-19 pandemic and mood swings [6], it seems that these effects may be more in people who did collaborative in team sports compare to individual sports. Research has shown that people who do individual sports have different self-regulatory skills [7], coping strategies [8], some individual characteristics [9,10], and personality characters [11] from people who do team sports. Additionally, it has been claimed that team sport athletes are at high genetic risk of severe COVID-19 [12]. Therefore, the second aim of this study was the investigation of individual and team sport athlete’s mood states during the COVID-19 pandemic. For a deeper understanding of the issue, we investigated contact and non-contract individual and team sport athletes.

In addition to the study of exercise frequency and intensity, we also predicted them by other influential factors. It seems that craving for doing a sport, which is known as exercise dependency, along with individual factors such as age and gender, are some possible predictors of exercise indexes during the COVID-19 pandemic. Therefore, the third aim of this study was predicting exercise frequency and intensity by individual characteristics including exercise dependency, age, gender, and adherence to quarantine.

Although the positive relationship between higher expert-paced PA intensity levels and mood states have been shown [13,14,15], it seems that a lockdown-related change in PA levels is associated with mood, which, in turn, is influenced by the type of physical activity (team or individual sport) and exercise dependency in individuals. As previous studies have shown [16], there seemed to be a conceptual relationship between mood states during the COVID-19 pandemic and exercise dependency in this study; therefore, we investigated possible relationships between them among all groups.

Mental health conditions among the general and professional populations were reported by cross-sectional [17] and longitudinal [18] studies during the COVID-19 pandemic, but there currently is not clear evidence on the possible effect of the dependence of exercise on the other athlete’s behavior which, in turn, could affect PA. Therefore, in this study, we investigated non-contact individual sport athletes (NCISA), contact individual sport athletes (CISA), and team sport athletes (TSA) in terms of exercise indexes, exercise dependency, and mood states.

To the best of our knowledge, no study to date has evaluated the differences in frequency and intensity of PA, exercise dependency and positive/negative mood states, and their possible differences among individual and team sport athletes during the COVID-19 pandemic. Therefore, we wanted to find out the difference in exercise characteristics and mood states among contact and non-contract individual and team sport athletes. Additionally, we investigated the possible relationship between exercise dependency and positive/negative mood states. Finally, we looked to find out whether exercise frequency and intensity could be predicted by exercise dependency, age, gender, and adherence to quarantine during the COVID-19 pandemic.

## 2. Materials and Methods

### 2.1. Procedure

Individual and team sports athletes were approached via social network sites (SNS) to participate in the present online study, and they were asked to fill out a questionnaire package on exercise frequency and intensity before and during the COVID-19 pandemic, mood states, and exercise dependency from 5 March to 30 April 2020. Before starting, the objectives of the research, the anonymous data gathering techniques, the confidential data handling practices, and the ethical approval of the study were explained to the participants on the first pages of the study. Next, participants accepted informed consent by clicking a box of agreement. Additionally, the Human Research Ethics Board at the Sport Sciences Research Institute of Iran approved the study (approval ID: IR.SSRC.REC.1399.070), which was performed in accordance with the last revision of the Declaration of Helsinki [19].

### 2.2. Participants

A sample of 1353 Iranian athletes with 45.4% females participated in this study. They included non-contact individual sportspeople (skating, *n* = 95; fitness and body building, *n* = 165; swimming, *n* = 80; gymnastics, *n* = 70) with athletes whose mean age was 26.8 years (SD = 10.53 years); contact individual sport athletes (karate, *n* = 85; taekwondo, *n* = 87; judo, *n* = 94, wushu, *n* = 76; boxing, *n* = 91; wrestling, *n* = 68) whose mean age was 23.76 years (SD = 9.86 years); and team sport athletes whose mean age was 24.79 years (SD = 10.41 years) (football, *n* = 102; futsal, *n* = 85; volleyball, *n* = 98; handball, *n* = 115; and basketball, *n* = 110).

### 2.3. Measures

#### 2.3.1. Exercise Level 

Exercise levels were measured by inquiring about the type, frequency, and intensity of exercise (from low to very high) before and during the COVID-19 pandemic, which was extracted from the 5-item PA questionnaire developed based on Cho’s study [20]. The reliability and validity of this tool have been confirmed by Cho [21]. The first question was related to the type of activities in which the athletes participated before/during the COVID-19 pandemic. Individual and team sports were the main sports of athletes before the COVID-19 pandemic; an open-ended question was asked about physical activities during the COVID-19 pandemic. The second question was “before/during the COVID-19 pandemic, how often do you participate in the activity?” The choices were “every day, 6 days/week, 5 days/week, 4 days/week, 3 days/week, 2 days/week, 1 day/week and anytime”. Additionally, the last question was “how intensely do you participate in the activity before/during the COVID-19 pandemic?” The choices for intensity were “light, moderate hard and very hard”.

#### 2.3.2. Mood State

To evaluate positive and negative mood states, we used a shortened version of the Brunel Mood Scale (BRUMS) [22,23]. The questionnaire included items related to 16 mood states. In the mood test, the participants were asked to express their current feelings according to the instructions. Each response was scored on a five-point scale (ranging from 0 = no to 4 = extremely). The internal consistency values (Cronbach’s alpha) of all dimensions and the total scale ranged from 0.82 to 0.96 [24], while in the present study the total scale was 0.90.

#### 2.3.3. Exercise Dependency

The Exercise Dependency measure was measured via 16 items on a seven-point Likert scale. It included the following five factors: expected positive consequences, interference with social life, health, withdrawal symptoms, and exercise as a possibility to compensate for psychological problems. This scale has already been used by previous researchers and validated with internal consistency (α = 0.643–0.808) and fitted the model [25]. 

Additionally, questions regarding some individual characteristics such as age and social measures concerning the COVID-19 pandemic were asked, such as adherence to quarantine, type and duration of applied confinements, social distancing, and lockdown of gyms, outdoor sports centers and parks.

### 2.4. Statistical Analysis

Differences in exercise indexes among athletes of different sports were analyzed by Kruskal–Wallis one-way analysis of variance for between-group effects, and Mann–Whitney U test for within-group effects. Additionally, multivariate analysis of variance was used to investigate negative and positive mood states among different sport groups during the COVID-19 pandemic. In addition, the relationship of exercise dependency with mood states was analyzed by Spearman’s correlation coefficients. Additionally, ordinal regressions were used to predicting exercise frequency and intensity by exercise dependency, age, gender, and adherence to quarantine among different sport groups. Finally, one-way analysis of variance and MANOVA were used to analyze exercise dependency and its subscales among different sport groups. The level of significance was set at alpha < 0.05. All statistical analyses were computed utilizing IBM Corp. Released 2015. IBM SPSS Statistics for Windows, Version 23.0. Armonk, NY, USA: IBM Corp and Microsoft Excel (2013). 

## 3. Results

### 3.1. Descriptive Statistics of Studied Variables 

Table 1 shows the mean or median scores of exercise frequency and intensity before and during the COVID-19 pandemic, as well as mood states, exercise dependency, and their subscales.

### 3.2. Exercise Indexes of Different Sport Athletes 

Kruskal–Wallis one-way analysis of variance showed that NCISA had less frequent exercise than CISA, both before and during the COVID-19 pandemic, and NCISA had less frequent exercise than TSA during the COVID-19 pandemic. Regarding exercise intensity, CISA had higher scores than NCISA and TSA before the COVID-19 pandemic, and CISA had higher exercise intensity than TSA during the COVID-19 pandemic (Table 2). 

In addition, Mann–Whitney U test for within-group effects showed that the frequency and intensity were reduced from before to during the COVID-19 pandemic in the three groups, except for TSA intensity (Table 1 and Table 3). However, this latter variable was not significant.

### 3.3. Mood States of Athletes during the COVID-19 Pandemic

Descriptive analyses of mood states (12 items for negative mood states and 4 items for positive mood states) among NCISA, CISA, and TSA are shown in Figure 1.

MANOVA showed that there were significant differences in negative mood states among different athletes (F = 1.66, *p* = 0.022, Wilks’ Lambda = 0.968, Partial Eta Squared = 0.016). Thus, the pairwise comparisons showed that CISA were more discouraged than NCISA (mean differences = 0.253, *p* = 0.010). Additionally, positive mood state analysis showed that there were significant differences among different sports (F = 2.10, *p* = 0.032, Wilks’ Lambda = 0.986, Partial Eta Squared = 0.007). The results showed that CISA were more vigorous than TSA (mean differences = 0.243, *p* = 0.005). However, there were no significant differences among other negative or positive mood states (all *p*-values > 0.05). 

### 3.4. Correlation Coefficients of Exercise Dependency, Negative and Positive Mood States 

The Pearson’s correlation coefficients also showed that there was a positive significant relationship between exercise dependency with positive mood states (r = 0.198, *p* = 0.0001) and a negative significant correlation with negative mood states (r = −0.077, *p* = 0.008), but their effect size was very small (4% and 0.5%, respectively). 

### 3.5. Predicting of Exercise Indexes by Exercise Dependency, Age, Gender, and Adherence to Quarantine in Different Sport Groups 

Additionally, we investigated whether exercise frequency and intensity could be predicted by exercise dependency, age, gender, and adherence to quarantine among different sports. For NCISA, CISA, and TSA, ordinal regressions showed that adherence to quarantine and exercise dependency were the best predictors. (Table 4). 

Parameter estimates showed that exercise frequency could be predicted by exercise dependency (in the NCISA and CISA), adherence to quarantine (in the CISA and TSA), and age (in the CISA). In addition, exercise intensity could be predicted by exercise dependency (CISA), adherence to quarantine (in the NCISA and CISA), and gender (in the CISA). 

### 3.6. Perceived Exercise Dependency and Its Subscales among Sports Groups during COVID-19

Finally, one-way analysis of variance showed that there was no significant difference in exercise dependency (total score) among different sport groups (F = 2.08, *p* = 0.132). However, MANOVA (Wilk’s Lambda value = 0.978, F = 2.64, *p* = 0.003) showed that CISA had more withdrawal symptoms than NCISA and TSA (mean differences = 0.75, 1.17; *p* = 0.049, 0.002, respectively). Additionally, regarding health status, NCISA had higher scores than TSA (mean differences = 0.94, *p* = 0.006).

## 4. Discussion

This survey reports some data from online research among Iranian individual and team sport athletes during the COVID-19 pandemic. A total of 1353 athletes from individual and team sports participated in this study. Changes in exercise before and during the COVID-19 pandemic showed that the intensity and frequency of exercise were higher before the COVID-19 pandemic than during it among all groups, apart from the intensity metric of TSA. Interestingly, intensity of TSA was higher during the COVID-19 pandemic but was not significant. These results are in line with most studies among the general population [5,26,27,28]. The scientific community has highlighted the real benefits of PA during the pandemic [29,30]; however, our results showed a reduction in exercise indexes among NCISA and CISA and exercise frequency among TSA. It seems that most team athletes may have performed home exercise, fitness, stretching, walking, and individual exercises at different intensities during this pandemic. Although the frequency of exercise decreased among them, in-depth checking of the present study results showed that the type of exercises had changed, which complied with exercise and PA recommendations during the coronavirus outbreak [31]. This experience could be due to essential changes in sports training schedules. Sports training was discontinued at the original coronavirus outbreak and research data were collected at that time; therefore, the present findings can be justified. This issue was consistent with Lim [32], who cited that many athletes attempt to maintain active lifestyles by themselves. Therefore, changes in daily life activities are necessary. It seems TSA may push themselves to keep fit and stay in shape by doing some other types of PA and exercise. It should also be noted that at the time of data collection of the present study, all places of sports activities, both individual and team, were closed, except for unorganized individual activities outdoors. However, at the same time, there were no nationwide closures for parks, shopping malls, etc., which may have affected the obtained results. 

To address this issue, Mutz and Gerke [33] reported a significant decline in sport and exercise activities among Germans. Additionally, about one-third of the studied population reduced their sport and exercise activities, and only 6% intensified sport and exercise levels. They cited that this last group increased home-based workouts and outdoor endurance sports, while others did not adapt their sporting routines to the present conditions.

Based on Figure 1, all the positive moods were less than the negative ones among all groups. However, our results showed that CISA were more discouraged and vigorous. It seems that the type of sport could affect negative and positive mood states; sports in which the participants necessarily come into bodily contact with one another seemed to be more affected. Results of negative moods are consistent with recent research which indicates that the increasing menace of the epidemic has resulted in depression due to disrupted travel plans, social isolation, and media information overload [34].

In addition, positive relationships of exercise dependency with positive mood states and negative relationships with negative mood states were in line with a previous study [16]. They showed small to moderate correlations between exercise dependence with mood states. Previous research suggested that mood states could play a critical role in the development or the maintenance of exercise dependency [16,35,36]. The “affect regulation” hypothesis could justify this issue [36]. Therefore, PA results in improvements in positive mood states and decreases in negative mood states. As the exercise cycle continues, increased amounts of exercise are needed to experience improvement in affect and mood. On the other hand, lack of exercise can lead to a weakening of positive moods and an increase in negative moods.

In addition, exercise indexes were predicted by age, gender, adherence to quarantine, and exercise dependency in the three groups. Adherence to quarantine and exercise dependency were the best predictors of exercise frequency and intensity. Additionally, age and gender were able to predict the frequency and intensity of exercise in the CISA group. 

It was further shown that CISA had unpleasant feelings of leaving exercise, and TSA had an unhealthier status than NCISA. Therefore, it seems that the nature of how an athlete interacts with other athletes may be important in the context of athletes’ feelings during the COVID-19 pandemic.

Despite the new findings, several limitations warn against overgeneralization of the results. Firstly, the cross-sectional design of the study precludes conclusions about the studied variables. Secondly, although we used open- and closed-ended questions in this study, data collection through self-reporting may be biased. Thirdly, the present study data were collected only a few months after the onset of the COVID-19 pandemic; however, longitudinal studies seem to better clarify the changing trends of exercise indexes. 

## 5. Conclusions

Among a large sample of Iranian athletes of different ages, gender, and sports, changing exercise indexes were not similar among groups; there was a dominant reduction pattern among all sports, and a non-significant increasing trend was also observed in team sports. Unlike previous studies, the present project did not focus only on the overall scores of negative or positive mood; it also presented new findings related to negative and positive mood subscales. Additionally, in this study, we showed that exercise dependency has a significant relationship with both positive and negative mood states. Finally, it was shown that with increasing exercise dependency, exercise intensity and frequency increased. Additionally, adherence to quarantine and exercise dependency were the best predictors of exercise indexes during the COVID-19 pandemic. 

## Figures and Tables

**Figure 1 children-08-00438-f001:**
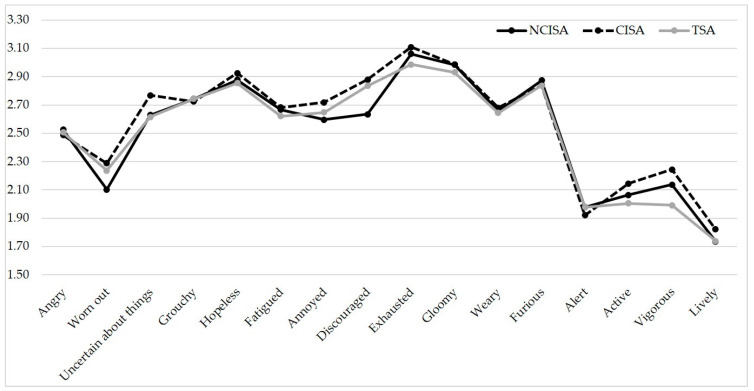
Mood states of different groups of athletes during the COVID-19 pandemic.

**Table 1 children-08-00438-t001:** Descriptive statistics of studied variables.

Variables		NCISA	CISA	TSA
		Mean/Median	Mean/Median	Mean/Median
Before COVID-19	Frequency	3.87	4.29	4.30
	Intensity	2.68	3.00	2.66
During COVID-19	Frequency	2.52	3.14	4.65
	Intensity	1.93	2.03	1.87
Mood state (Total score)		39.17	40.67	39.68
Positive moods		7.70	8.03	7.62
Negative moods		31.4	32.64	32.05
Exercise dependency (Total score)		66.68	67.02	64.67
Expected positive consequences		17.04	17.05	16.45
Withdrawal symptoms		13.54	14.27	13.14
Exercise as a possibility to compensate for psychological problems		11.21	11.59	11.13
Interference with social life		7.44	7.28	7.63
Health		16.75	16.14	15.70

**Table 2 children-08-00438-t002:** Group differences of exercise frequency and intensity before and during COVID-19.

			Frequency				Intensity			
Exercise	(I)	(J)	Kruskal Wallis	*p*	Test Statistics	Adj. *P*	Kruskal Wallis	*p*	Test Statistics	Adj. *P*
Before COVID-19	NCISA	CISA	7.27	0.026 *	−65.28	0.034 *	58.71	0.0001 *	−152.53	0.000 *
		TSA			−55.99	0.096			4.43	0.999
	CISA	TSA			9.29	0.99			156.97	0.000 *
During COVID-19	NCISA	CISA	130.86	0.000 *	−91.24	0.001 *	9.30	0.01 *	−42.08	0.216
		TSA			−291.91	0.000 *			26.37	0.806
	CISA	TSA			−200.67	0.000 *			68.45	0.008 *

NCISA, non-contact individual sport athletes; CISA, contact individual sport athletes; TSA, team sport athletes; * *p* ≤ 0.05.

**Table 3 children-08-00438-t003:** Intergroup changes in exercise frequency and intensity from before to during COVID-19.

			Frequency		Intensity	
Group	(I)	(J)	Wilcoxon Statistics	*p*	Wilcoxon Statistics	*p*
NCISA	Before COVID-19	During COVID-19	8275.50	0.000 *	2051.00	0.000 *
CISA	Before COVID-19	During COVID-19	13,884	0.000 *	2128.50	0.000 *
TSA	Before COVID-19	During COVID-19	32,120	0.0052	3228.00	0.000 *

NCISA, non-contact individual sport athletes; CISA, contact individual sport athletes; TSA, team sport athletes; * *p* ≤ 0.05.

**Table 4 children-08-00438-t004:** Ordinal regression parameters of NCISA, CISA, and TSA during the COVID-19 pandemic.

		Model Fit			Goodness-of-Fit		
	Sport Type	−2 Log Likelihood	Chi-Squared	*p*	Pearson’s Chi-Squared	*p*	Pseudo R-Square
Frequency	NCISA	1274.11	189.63	0.000 *	3087.41	0.000 *	0.41
	CISA	1497.14	168.77	0.005 *	5129.439	0.000 *	0.34
	TSA	1446.91	184.21	0.000 *	3266.846	0.015 *	0.36
Intensity	NCISA	505.69	255.54	0.000 *	936.560	0.005 *	0.58
	CISA	678.490	234.07	0.000 *	1361.711	0.000 *	0.48
	TSA	650.81	174.51	0.001 *	1057.76	0.226	0.40

NCISA, non-contact individual sport athletes; CISA, contact individual sport athletes; TSA, team sport athletes; * *p* ≤ 0.05.

## Data Availability

The data presented in this study are available on request from the corresponding author.

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
