# Peer review of "Different Effects of the COVID-19 Pandemic on Exercise Indexes and Mood States Based on Sport Types, Exercise Dependency and Individual Characteristics"

_children, 2021, doi:10.3390/children8060438_

Round 1

Reviewer 1 Report

This study analysed the results of a cross-sectional survey of Iranian Athletes during the COVID-19 pandemic. The study design is interesting and there is a wide variety of athletes included in the study. 

Major Comments:

In the abstract line 18/19 states that exercise frequency, intensity, dependency and mood states were examined before and during the COVID-19 pandemic but the survey examined current mood and exercise dependency and only examined changes in frequency and intensity before and during COVID-19. 

It would be important to give the context of restrictions on the athletes when they completed the survey - was there a lockdown in place? How long was the survey left open - where there any changes in restrictions during the time the survey was being answered? 

The change in frequency of sport was interesting and how this differed between the NCSIA and TSA. I would like to have seen a table of descriptive statistics (Table 1) which shows the level of frequency and intensity before and during COVID and also shows summary measures for Exercise dependency and Mood. 

Exercise level: For exercise frequency is it measured as the number of days per week that the athlete participates in sport/training? How exactly is it coded and treated as a quantitative variable? 1 = 1 day, 2 = 2 days etc? 

Exercise Intensity is an ordinal categorical variable and is not suitable for being a dependent variable in a regression analysis. 

Statistical Analysis: 

Repeated measures ANOVA may be suitable for exercise frequency (if the data is approximately normally distributed) but it is not appropriate for intensity as it is an ordinal categorical variable with only 3 levels. 

For figure 2 (Mood states mean scores) - I'm not sure what information is given by this graph? Maybe there is a better way of displaying this information or it might be enough to focus the analysis on the Total, positive and negative mood scores.

MANOVA analysis is suitable for differences in multiple dependent quantitative variables. For Mood you have Total scores and subscales of positive and negative scores. MANOVA is not needed here if you are only looking at the differences in one variable (eg Total Mood) between groups (TSA, NCISA and CISA) - ANOVA would be sufficient. I'm not sure why the scores give on Figure 2 are higher for positive and negative moods than the total mood score - more information on how these variables were scored would be valuable. Also no decimal places are required in Figure 2 on the vertical axis. 

The predictive models presented in Tables 3 and 4 have very low R2 values. Intensity as a categorical variable should not be analysed using a linear regression model. I don't see anything in these predictive models that gives new information. 

While there are interesting aspects to the data eg how TSA, CISA and NCISA athletes were affected differently by the COVID pandemic in Iran, the analysis needs to be more careful considered to bring this information to the reader. 

Minor Comments: Throughout the manuscript there were grammatical errors and typos - these would need to be corrected. 

Author Response

Dear Editor, 

Thank you!

Reviewer 2 Report

Dear authors,
The submitted manuscript has very good intentions and although it is a topic of interest to the scientific community, I observe technical inconsistencies and some weaknesses. I comment on them below:

-Some questions to answer are: What is the main question addressed by the research? Is it relevant and interesting?
-What does it contribute to the subject area compared to other published materials?
-Are the conclusions consistent with the evidence and arguments presented? Do they address the main question posed?
-The methodology used does not have sufficient consistency to be published in a high impact journal.
-The results do not have scientific consistency to be published in a high impact journal. Indicate how the results obtained are related or not to the literature and the opinion of prominent authors on the subject.
- Connect the conclusions in relation to the results obtained so that they present greater forcefulness.
- Consistent conclusions must be included that confront the results obtained with the experts in the field.
- The discussion and conclusions are not related to the main objective.
- It is recommended to update the bibliography.

Author Response

Dear Editor, 

Thank you!

Round 2

Reviewer 2 Report

Dear editor,

The submitted manuscript has improved with the latest corrections so my decision is to admit its publication. Congratulations!

Kind regars,

This manuscript is a resubmission of an earlier submission. The following is a list of the peer review reports and author responses from that submission.